# Inhibitory Effects of Menadione on *Helicobacter pylori* Growth and *Helicobacter pylori*-Induced Inflammation via NF-κB Inhibition

**DOI:** 10.3390/ijms20051169

**Published:** 2019-03-07

**Authors:** Min Ho Lee, Ji Yeong Yang, Yoonjung Cho, Hyun Jun Woo, Hye Jin Kwon, Do Hyun Kim, Min Park, Cheol Moon, Min Ji Yeon, Hyun Woo Kim, Woo-Duck Seo, Sa-Hyun Kim, Jong-Bae Kim

**Affiliations:** 1Department of Biomedical Laboratory Science, College of Health Sciences, Yonsei University, Wonju 26493, Korea; lmh77777@naver.com (M.H.L.); clever1088@nate.com (J.Y.Y.); haejin5462@naver.com (H.J.K.); rlaehgus00@naver.com (D.H.K.); amaranth1001@nate.com (H.W.K.); 2Forensic DNA Division, National Forensic Service, Wonju 26460, Korea; yoonjungs@korea.kr; 3Department of Clinical Laboratory Science, College of Medical Sciences, Daegu Haany University, Gyeongsan 38610, Korea; taesube@nate.com; 4Department of Biomedical Laboratory Science, Daekyeung University, Gyeongsan 38547, Korea; pm@tk.ac.kr; 5Department of Clinical Laboratory Science, Semyung University, Jecheon 27136, Korea; moonc72@naver.com; 6Natural Products Research Center, Korea Institute of Science and Technology (KIST) Gangneung 25451, Korea; mkkk01@naver.com; 7National Institute of Crop Science (NICS), Rural Development Administration (RDA), Wanju-Gun 55365, Korea; swd@rda.go.kr

**Keywords:** *H. pylori*, menadione, CagA, VacA, T4SS, inflammation, NF-κB, IL-8

## Abstract

*H. pylori* is classified as a group I carcinogen by WHO because of its involvement in gastric cancer development. Several reports have suggested anti-bacterial effects of menadione, although the effect of menadione on major virulence factors of *H. pylori* and *H. pylori*-induced inflammation is yet to be elucidated. In this study, therefore, we demonstrated that menadione has anti-*H. pylori* and anti-inflammatory effects. Menadione inhibited growth of *H. pylori* reference strains and clinical isolates. Menadione reduced expression of *vac*A in *H. pylori*, and translocation of VacA protein into AGS (gastric adenocarcinoma cell) was also decreased by menadione treatment. This result was concordant with decreased apoptosis in AGS cells infected with *H. pylori*. Moreover, cytotoxin-associated protein A (CagA) translocation into *H. pylori*-infected AGS cells was also decreased by menadione. Menadione inhibited expression of several type IV secretion system (T4SS) components, including *vir*B2, *vir*B7, *vir*B8, and *vir*B10, that are responsible for translocation of CagA into host cells. In particular, menadione inhibited nuclear factor kappa-light-chain-enhancer of activated B cell (NF-κB) activation and thereby reduced expression of the proinflammatory cytokines such as IL-1β, IL-6, IL-8, and TNF-α in AGS as well as in THP-1 (monocytic leukemia cell) cell lines. Collectively, these results suggest the anti-bacterial and anti-inflammatory effects of menadione against *H. pylori*.

## 1. Introduction

*Helicobacter pylori* (*H. pylori*) is a Gram-negative bacterium possessing a characteristic helical appearance. *H. pylori* primarily colonizes in the human stomach and has been reported to infect approximately half of the world population [1]. Infection of *H. pylori* on gastric mucosa is associated with various gastric diseases including inflammation, chronic gastritis, peptic ulcers, and gastric adenocarcinoma [2]. The World Health Organization classified *H. pylori* as a class I carcinogen because of its involvement in gastric cancer development [3]. It is estimated that more than half of the adult population is infected with *H. pylori* worldwide, and its infection is responsible for 75% of all gastric cancer cases [4]. Therefore, a concerted effort for eradication of *H. pylori* infection is necessary for health promotion worldwide.

The most studied virulence factors of *H. pylori* are cytotoxin-associated protein A (CagA) and vacuolating cytotoxin A (VacA). CagA translocates to the host cells by the type IV secretion system (T4SS) [5]. The T4SS of *H. pylori* consists of up to 32 proteins, but many of their functions are yet unknown [6]. However, the functions of 11 VirB proteins (VirB1–11) and a VirD4 protein have been studied based on their homology to the agrobacterial T4SS proteins [6,7,8,9]. The VirB and VirD4 proteins assemble to form three subparts consisting of a cytoplasmic/inner membrane complex (VirB4, VirB6, VirB8, VirB11, and VirD4), a double membrane-spanning channel (VirB7, VirB9, and VirB10), and an external pilus (VirB2 and VirB5), and these three subparts are interlinked [6,7]. Kwok et al. demonstrated that α_5_β_1_ integrin is a host cell receptor that directly binds to VirB5 (CagL) proteins of *H. pylori* [6]. Once CagA is injected, host cell Src kinases phosphorylate the EPIYA motif of CagA proteins and subsequently deregulate intracellular signaling transduction pathways, disrupt epithelial cell junctions, and induce inflammation [10,11,12,13]. VacA has been known to induce cytoplasmic vacuole formation [14]. VacA protein secretion is associated with the type Va system [14,15,16]. Translocation across the inner-membrane is mediated by Sec-related proteins [15,16]. The signal peptide region of the VacA protein is recognized by SecYEG for the translocation through the inner-membrane [15,16]. SecA is an especially important regulatory protein because it is an ATPase that provides energy necessary for translocation of the proteins by Sec-related proteins [15,16,17]. VacA, which translocates to the host cells, interacts with host cell mitochondria, resulting in apoptosis via activation of the intrinsic caspase cascade [18,19,20,21,22]. 

One of the mechanisms by which *H. pylori* infection progresses to gastric carcinogenesis is the persistent presence of the pathogen, which leads to the development of chronic inflammation accompanied by infiltration of neutrophils and lymphocytes as well as the production of proinflammatory cytokines [23]. The gastric mucosal levels of the proinflammatory cytokines are increased in *H. pylori*-infected subjects, and these include IL-1β, IL-6, IL-8, and TNF-α [24,25]. Especially, *H. pylori*-mediated IL-8 secretion in gastric epithelial cells requires activation of the nuclear factor kappa-light-chain-enhancer of activated B cells (NF-κB), while *H. pylori* strains that fail to induce IL-8 secretion do not activate NF-κB [26,27]. In the absence of an activating stimuli, NF-κB remains inactive in the cytoplasm bound to a family of inhibitory proteins known as inhibitors of NF-κB (IκBs). Activated NF-κB forms a homo- or heterodimer and translocates to the nucleus to function as a transcription factor [28]. In particular, NF-κB is clearly one of the most important regulators for expression of proinflammatory cytokines [29]. Activation of NF-κB by *H. pylori* induces nuclear translocation, which causes increased expression of NF-κB responsive genes including TNF-α, IL-1β, IL-6, and IL-8 [27]. NF-κB activation is also known to regulate cellular growth responses including apoptosis and is required for the induction of inflammatory and tissue-repair genes [23,27,30].

Menadione (2-methyl-1,4-naphthoquinone) is a synthetic form of vitamin K. It is also called vitamin K3. Menadione has a higher anti-hemorrhagic activity than the naturally occurring vitamin K (VitK1 and VitK2) [31]. The generally known roles of vitamin K are the maintenance of blood clotting and bone formation [31,32]. There are several reports demonstrating the anti-bacterial effect of menadione [33,34,35,36]. Andrade et al. reported the antibiotic-modifying activity of menadione in multi-resistant strains of *Staphylococcus aureus*, *Pseudomonas aeruginosa*, and *Escherichia coli*. They suggested that menadione potentiated aminoglycosides against multi-resistant bacteria since it lowered the minimal inhibitory concentration (MIC) of the antibiotics tested in their experiments [37]. In particular, Park et al. described the inhibitory effect of menadione against *H. pylori* in the screening of various naphthoquinones by a disk diffusion assay [34].

Antibiotic resistance of *H. pylori* is continuously increasing, and the development of a new therapeutic agent to support *H. pylori* treatment is necessary. Menadione was reported to have anti-bacterial activity. Therefore, an inhibitory effect of menadione on *H. pylori* growth and the effects of menadione on CagA and VacA, major virulence factors of *H. pylori,* were investigated in this study. Furthermore, menadione inhibited NF-κB activation and possessed anti-inflammatory activity according to the previous reports [38]. Thus, the effects of menadione on the expression of the inflammatory cytokines and the NF-κB-mediated signaling pathway during *H. pylori*-induced inflammatory response in vitro was investigated. 

## 2. Results

### 2.1. The Inhibitory Effect of Menadione on the Growth of H. pylori

An agar dilution test was performed to determine the MIC of menadione against *H. pylori*. Mueller–Hinton agar containing various concentrations of menadione was prepared and *H. pylori* reference strains (ATCC 49503, ATCC 26695, SS1, and HP51) were grown on agar plates for 72 h. According to the agar dilution test, the MIC of menadione against *H. pylori* was 8 μM (Figure 1). Clinical isolates of *H. pylori* were collected from gastric biopsies, and the MIC of menadione was determined to confirm whether menadione can inhibit growth of *H. pylori* clinical isolates as well as the reference strains. Among the 38 clinical isolates, the MIC of 57.9% (22/38) was 8 μM, 21.1% (8/38) was 4 μM, and 10.5% (4/38) was 2 μM (Table 1). These results showed that menadione has an anti-bacterial effect on the clinical isolates of *H. pylori* as well as the reference strains.

### 2.2. Menadione Reduced CagA and VacA Translocation to AGS Cells and Recovered Morphological Changes Caused by H. pylori Infection

CagA and VacA proteins are the most well-known cytotoxic proteins secreted by *H. pylori*. A notable feature appearing after CagA translocation to host cells is the rearrangement of the host cell actin cytoskeleton leading to the change of cell morphology, the so-called hummingbird phenotype [39]. The VacA protein internalizes into the host cell and leads to vacuolation, which is characterized by the accumulation of large vesicles [14]. In the experiment, it was also found that the morphological changes in the *H. pylori*-infected AGS cells were alleviated by menadione treatment in a dose-dependent manner (Figure 2A). It was presumed that the translocation of the CagA and VacA proteins to AGS cells may have decreased by menadione treatment. Thus, the *H. pylori-* and menadione-treated AGS cells were harvested and subjected to Western blotting to investigate protein levels of the CagA and VacA in the AGS cells. The CagA and VacA proteins were translocated to the infected AGS cells, and translocation of both proteins was decreased by menadione treatment as predicted (Figure 2B). In particular, the CagA protein level dramatically decreased even by a low dose of menadione treatment (Figure 2B,C). These results indicated that menadione reduced translocation of CagA and VacA proteins to AGS cells and inhibited *H. pylori*-induced morphological changes (hummingbird phenotype and vacuolations) of AGS cells. To investigate whether menadione directly reduced the expression of CagA and VacA in the bacteria, the mRNA levels of CagA and VacA in *H. pylori* treated with menadione were assessed by RT-PCR. Menadione reduced the expression of VacA, but it did not reduce the expression of CagA in *H. pylori* (Figure 3). 

CagA and VacA proteins are secreted by type IV and type V secretion systems, respectively. Therefore, whether menadione affected the secretion system of *H. pylori* was investigated. Specific primers were designed for targeting components of the secretion systems, and the expressions were examined using RT-PCR. In the result, mRNA expressions of *vir*B2, *vir*B7, *vir*B8, and *vir*B10 were decreased by menadione treatment (Figure 4A,B). Moreover, RT-PCR was performed targeting for integrin α_5_β_1_, which acts as a receptor for the T4SS on the host cell surface, but expression of this integrin was not changed (Figure 4C). RT-PCR was performed to investigate whether menadione has an effect on the mRNA expression of *sec*A, which is a regulator of the T5SS. However, no change was observed in this experiment regarding the expression of *sec*A (Figure 4D). Collectively, these data indicate that menadione reduced the translocation of VacA to the host cells by the downregulation of *vac*A mRNA expression in *H. pylori*, and reduced the translocation of CagA by the downregulation of T4SS components necessary for CagA secretion. By inhibiting the translocation of CagA and VacA proteins, menadione alleviated morphological changes of AGS cells induced by *H. pylori* infection.

### 2.3. Menadione Alleviated H. pylori-Induced Apoptosis and Death of AGS Cells

*H. pylori* infection induces apoptosis of gastric epithelial cells [19,20,21,22]. In this study, it was investigated whether menadione can inhibit apoptosis induced by *H. pylori* in AGS cells. AGS cells were infected with *H. pylori* and treated with menadione for 48 h. After incubation, the cells were observed by an inverted microscope. It was found that cell confluency was decreased by *H. pylori* infection, but the decrease of cell confluency was inhibited by menadione treatment in a dose-dependent manner (Figure 2A). Additionally, cell viability was measured by a WST assay, and it was found that *H. pylori* infection (200 MOI) reduced cell viability of AGS cells (46.57%) but that the cell death was partially inhibited by menadione treatment (5 μM) (Figure 5A). The dose of menadione used in this experiment was enough to inhibit *H. pylori* growth without inhibiting the growth of AGS cells. This result showed that menadione rescued AGS cells from *H. pylori*-induced cell death. Since menadione reduced the translocation of VacA proteins in AGS cells in this study, it was hypothesized that menadione may inhibit *H. pylori*-induced apoptosis in gastric epithelial cells. In the Western blotting result, poly (ADP-ribose) polymerase (PARP) was cleaved by *H. pylori* infection, but it was inhibited by menadione treatment (Figure 5B,C). Menadione alone had no effect on the level of PARP in the range of concentration (Appendix A). Based on these results, it could be inferred that menadione inhibited *H. pylori*-induced apoptotic cell death in AGS cells by inhibiting VacA translocation and *H. pylori* growth. 

### 2.4. Menadione Reduces Expression of Inflammatory Cytokines Induced by H. pylori-Induced Infection via Inhibition of NF-κB Activation

It has been reported that *H. pylori* activates NF-κB, which is one of the most important regulators for expression of proinflammatory cytokines [27,29]. To investigate whether menadione inhibits *H. pyori*-induced activation of NF-κB and inflammatory responses, AGS cells infected with *H. pylori* were treated with menadione, and Western blotting was then conducted to detect IκBα. *H. pylori* infection reduced the IκBα protein level, and menadione treatment partially recovered the reduced IκBα protein level (Figure 6). This result indirectly indicates that menadione reduced NF-κB activation by *H. pylori*. Activated NF-κB translocates to the nucleus, so nuclear translocation of NF-κB should be observed to precisely investigate the activation of NF-κB. Protein levels of NF-κB in cytosol and the nucleus were measured to confirm the inhibitory effect of menadione on NF-κB activation. AGS cells were infected with *H. pylori* and treated with menadione, and the cell lysates were then separated into cytosol fraction and nuclear fraction. Western blotting results for NF-κB in each fraction showed that NF-κB in the nucleus was increased by *H. pylori* infection, but it was reduced by menadione treatment (Figure 7). AGS cells without *H. pylori* infections were treated with menadione, and there was no variation in IκBα and NF-κB (Appendix A). Furthermore, confocal microscopy shows that FITC-labeled NF-κB was increased in the nucleus after *H. pylori* infection but it was reduced by menadione treatment (Figure 8). These results have a confirmed inhibitory effect of menadione on NF-κB activation and nuclear translocation by *H. pylori*.

*H. pylori* infection increases the levels of the proinflammatory cytokines such as IL-1β, IL-6, IL-8, and TNF-α in gastric mucosa [24,25]. Especially, IL-8 secretion from gastric epithelial cells by *H. pylori* infection is closely associated with NF-κB activation [26,27]. Therefore, expression of the proinflammatory cytokines in the AGS cells was investigated after treatment with *H. pylori* and menadione. The RT-PCR results showed that IL-1β, IL-8, and TNF-α mRNA levels were increased by *H. pylori* infection, and menadione treatment reduced their expressions in AGS gastric epithelial cells (Figure 9A,C). Moreover, the anti-inflammatory effect of menadione was also investigated in a macrophage cell line THP-1. THP-1 monocytic leukemia cells were differentiated into macrophages by PMA treatment. The cells were then infected with *H. pylori* and treated with menadione. The RT-PCR results also showed a reduction in proinflammatory cytokine expressions by menadione treatment (Figure 9B,C). Collectively, these results indicate that menadione reduces the expression of proinflammatory cytokines that are induced by *H. pylori* infection in gastric epithelial cells and macrophages via inhibition of NF-κB activation and nuclear translocation.

## 3. Discussion

*H. pylori* infection is responsible for 75% of all the gastric cancer cases, and more than half of the adult population is infected with *H. pylori* worldwide [4]. Numerous reports have suggested the prevalence of clarithromycin resistance and the limitation of current first-line therapy [40,41,42]. Therefore, development of a new therapeutic or supportive agent to help the eradication of *H. pylori* is necessary.

In this study, the inhibitory effect of menadione against *H. pylori* growth and *H. pylori*-induced inflammation is elucidated. The MIC of menadione was demonstrated against *H. pylori* reference strains and clinical isolates. The MIC of menadione was 8 μM (1.38 μg/mL), which was similar to the MIC of the antibiotics clinically administered to treat *H. pylori* infection. In our previous report, we isolated 165 *H. pylori* clinical strains and evaluated the MIC of five antibiotics commonly used for the eradication of *H. pylori* [43]. The MICs of the clarithromycin, amoxicillin, tetracycline, metronidazole, and levofloxacin against the susceptible *H. pylori* strains were 0.008–0.125, 0.008–0.5, 0.031–2, 0.063–4, and 0.008–1 μg/mL [43]. 

Menadione inhibited the *H. pylori* growth via the downregulation of replication and transcription machinery of *H. pylori*. Menadione also downregulated virulence factors such as urease and VacA proteins, which are necessary for the successful colonization and pathogenesis of *H. pylori*. By using in vitro infection model, it was found that menadione reduced *H. pylori*-induced apoptosis and inflammation. In particular, menadione inhibited NF-κB activation, thereby reducing expression of proinflammatory cytokines.

The MIC of menadione against *H. pylori* was 8 μM (1.38 μg/mL) in the agar dilution test (Figure 1). Various phytochemicals and plant extracts, including panaxytriol [44], anacardic acids [45], six quinolone alkaloids including evocarpine [46], cabreuvin [47], and catechins [48], have been reported to inhibit *H. pylori* growth. The MIC of these molecules against *H. pylori* ranged from 10 to 200 μg/mL. Based on this result, the anti-*H. pylori* activity of menadione was more effective than those of other phytochemicals previously reported. Moreover, the anti-*H. pylori* activity of menadione was also demonstrated in the *H. pylori* clinical isolates (Table 1).

In the current study, menadione treatment reduced the expression of T4SS components as well as the translocation of CagA to AGS cells. The mRNA expressions of *vir*B2, *vir*B7, *vir*B8, and *vir*B10 in AGS cells were decreased by menadione treatment. Although *cag*A expression was not changed by menadione, decreased expression of the T4SS components can explain how menadione inhibited translocation of CagA protein into AGS cells. The periplasmic core of the T4SS consists of VirB8, VirB7, VirB9, and VirB10, and these proteins form a channel necessary for passenger proteins to pass through the periplasmic space. T4SS pili are generally composed of VirB2 and VirB5, and VirB2 is the major pilin subunit [7]. 

Inhibition of T4SS expression and CagA translocation was followed by the inhibition of NF-κB activation. Menadione treatment recovered the reduced IκBα protein level and inhibited the nuclear translocation of NF-κB in AGS cells infected with *H. pylori*, and this was also shown in the Western blotting and confocal microscopy results in this study. NF-κB exists in the cytoplasm in an inactivated form bound to a family of inhibitory proteins known as inhibitors of NF-κB (IκBs) [28]. Phosphorylation of IκB by IκB kinase (IKK) results in the ubiquitin-mediated degradation of IκB followed by activation and translocation of NF-κB to nucleus [28,29]. Therefore, these results collectively indicated the inhibitory effect of menadione on the NF-κB activation by *H. pylori*.

Chronic gastritis induced by *H. pylori* is characterized by the cagPAI-dependent expression of proinflammatory cytokines, which is largely mediated by the transcription factor NF-κB [49]. The transcription factor NF-κB regulates various genes that control the initiation of mucosal inflammatory responses induced by bacterial infection in human intestinal epithelial cells [50,51]. Activation of NF-κB by *H. pylori* induces nuclear translocation, which induces expression of NF-κB responsive genes including TNF-a, IL-1b, IL-6, and IL-8 [27]. Several reports have suggested the importance of CagA protein in the course of *H. pylori*-induced inflammation in infected gastric epithelial cells. Brandt et al. showed that CagA predominantly triggered NF-κB-mediated IL-8 production [52]. The Ras-Raf-Mek-Erk signaling cascade was involved in the CagA-induced NF-κB activation according to their study [52]. Lamb et al. also suggested the involvement of CagA in the activation of NF-κB. CagA contributed to the ubiquitination of TAK1 by TRAF6 and subsequently activated NF-κB [49]. These reports indicate that CagA acts as a multifunctional protein capable of triggering actin-cytoskeletal rearrangements, cell scattering by E-cadherin cleavage, and NF-κB activation and the production of IL-8 [49,52,53]. Furthermore, Viala et al. showed that *H. pylori* activates NF-κB via translocation of peptidoglycan by the T4SS in HEK293 cells. The translocation of peptidoglycan activated Nod1 and subsequently activated NF-κB. HEK293 cells do not have TLR2 or TLR4, so they are nonresponsive to LPS [54]. 

Subsequently, menadione treatment inhibited the expression of proinflammatory cytokines including IL-1β, IL-6, IL-8, and TNF-α induced by NF-κB. Several reports have suggested the anti-inflammatory effect of menadione [38,55,56,57]. Menadione inhibited concanavalin A-induced proliferation and cytokine production in lymphocytes and CD4 T cells. Menadione suppressed ERK, JNK, and NF-κB activation induced by concanavalin A in lymphocytes [56]. Menadione inhibited the TNF-α-induced nuclear translocation of NF-κB in HEK293 cells [38]. Menadione also inhibited the LPS-induced nuclear translocation of NF-κB and the production of TNF-α in a mouse macrophage cell line (RAW264.7) [38]. Menadione derivatives were also reported to have an anti-inflammatory effect on the mouse macrophage cell line, suppressing TNF-α production [55].

VacA causes alterations in the mitochondrial membrane, which changes transmembrane potential and results in the release of cytochrome *c* [21,58,59]. Released cytochrome *c* triggers an intrinsic apoptotic cascade [20,22]. Menadione reduced the translocation of VacA to AGS cells, which is correlated with the decrease of vacuolation in the cells. Moreover, inhibited PARP cleavage suggests that *H. pylori*-induced apoptosis was also reduced by menadione treatment, presumably by decreased VacA translocation. VacA was also implicated in the proinflammatory effect to immune cells. VacA stimulated production of TNF-α and IL-6 in mast cells, and stimulated COX2 expression in neutrophils and macrophages [14,60,61]. Therefore, it is presumed that decreased VacA translocation may also contribute to the anti-inflammatory effect of menadione.

In this study, we confirmed the anti-bacterial activities of menadione against *H. pylori*. Furthermore, we demonstrated that menadione had anti-inflammatory effects by decreasing the injection of virulence factors into host cells. Further studies are required to fully elucidate the anti-inflammatory and anti-bacterial mechanism of menadione against *H. pylori*. Previous studies that illuminated the anti-inflammatory effect of menadione used various types of cells including epithelial cells, macrophages, and T cells. Thus, it would be interesting to determine the anti-inflammatory effect of menadione during *H. pylori* infection in lymphocyte cell lines and primary immune cells. In addition, the use of animal models seems to be necessary to evaluate the safety and effectiveness of menadione in vivo.

## 4. Materials and Methods

### 4.1. Materials

The *H. pylori* reference strain ATCC 49503 was purchased from the American Type Culture Collection (ATCC, Manassas, VA, USA), and the *H. pylori* SS1 strain and the HP51 strain were obtained from *Helicobacter pylori* from the Korean Type Culture Collection, Gyeongsang National University (Jinju, Korea). Mueller–Hinton broth, Mueller–Hinton agar, and Brucella agar were purchased from Becton-Dickinson (Braintree, MA, USA) for cultivation of *H. pylori*. DMEM medium, fetal bovine serum, bovine serum, streptomycin-penicillin, and trypsin-EDTA were purchased from BRL Life Technologies (Grand Island, NY, USA) for mammalian cell culture. Trizol reagent, random hexamer, and Moloney murine leukemia virus reverse-transcriptase (MMLV-RT) were purchased from Invitrogen (Carlsbad, CA, USA) to perform RT-PCR. Menadione and protease inhibitor cocktail were obtained from Sigma-Aldrich (St. Louis, MO, USA). Antibodies to detect CagA, VacA, lamin B, and β-actin were purchased from Santa Cruz Biotechnology (Dallas, TX, USA), and the polyclonal antibody against whole *H. pylori* (ATCC 49503) was produced as previously described [27]. The antibody to detect GAPDH was purchased from Calbiochem (San Diego, CA, USA). Antibodies to detect PARP, NF-κB, and IκBα were purchased from Cell Signaling Technology (Danvers, MA, USA). The FITC-labeled anti-mouse antibody and ECL kit were purchased from Thermo Scientific (Waltham, MA, USA). DAPI stain solution was purchased from the Vector lab (Burlingame, CA, USA). The experiments were conducted with institutional biosafety committee approval (IBC No. 201705-P-001-01).

### 4.2. Collection of H. pylori Clinical Isolates

Gastric biopsy specimens for *H. pylori* isolation were collected at Yong-In Severance Hospital in Korea. The *H. pylori* strains were isolated from 38 patients undergoing gastroscopic examination to confirm the infection of *H. pylori*. Brucella agar plates supplemented with 10% bovine serum and *Helicobacter pylori* selective supplement (Oxoid Limited, Hampshire, UK) containing vancomycin, cefsulodin, trimethoprim, and amphotericin B were used as selective media. *H. pylori* was identified by colony morphology and urease test. The MICs were determined using a modified agar dilution method (Mueller–Hinton agar base containing 10% bovine serum) [62]. The experiments were conducted with institutional review board approval (IRB No. 1041849-201705-BR-056-01).

### 4.3. Agar Dilution Test to Determine MIC

*H. pylori* were grown on Brucella agar plates supplemented with 10% bovine serum at 37 °C for 72 h under humidified atmosphere with 5% CO_2_. *H. pylori* colonies grown on Brucella agar plates were collected and resuspended in saline. The number of bacterial particles in the *H. pylori* suspension was set to McFarland 2.0. Thirty microliters of the bacterial suspension were placed on the Mueller–Hinton agar supplemented with 10% bovine serum including indicated concentrations of menadione. The bacteria were incubated for 72 h, and MIC was determined based on the lowest concentration at which inhibition of growth was observed.

### 4.4. Mammalian Cell Culture

AGS gastric adenocarcinoma cells (ATCC CRL-1739) were cultured in DMEM medium supplemented with 10% fetal bovine serum and streptomycin-penicillin (100 μg/mL and 100 IU/mL). THP-1 monocytic leukemia cells (ATCC TIB-202) were cultured in RPMI 1640 medium supplemented with 10% fetal bovine serum and streptomycin-penicillin (100 μg/mL and 100 IU/mL). The differentiation of THP-1 monocytes into macrophages was conducted by treatment of 200 nM phorbol 12-myristate 13-acetate (PMA) for 48 h. All cells were incubated at 37 °C in a humidified atmosphere with 5% CO_2_.

### 4.5. RT-PCR

Cultured *H. pylori* or AGS cells were washed twice with phosphate-buffered saline (PBS) and total RNA was extracted using Trizol reagent as described in the manufacturer’s instructions. cDNA was synthesized by reverse transcription with 2 μg of total RNA, 0.25 μg of random hexamer and 200 U of MMLV-RT for 50 min at 37 °C and 15 min at 70 °C. Subsequent PCR amplification using 0.2 U of *Taq* polymerase was performed in a thermocycler using specific primers. The PCR primer sequences used in this study are listed in Table 2 [63,64,65,66,67]. 

### 4.6. Western Blotting

Cultured *H. pylori* or AGS cells were washed twice with PBS and then lysed with radioimmunoprecipitation assay (RIPA) buffer containing a protease inhibitor cocktail. The cell lysates were incubated on ice for 10 min. The cell lysates were then centrifuged and the supernatants were collected. The proteins were separated by SDS-polyacrylamide gel electrophoresis and transferred to a nitrocellulose membrane. The membrane was incubated with optimal concentrations of primary antibody (CagA, VacA, β-actin, PARP, IκBα, NFκB, Lamin B, or GAPDH) at 4 °C overnight and then incubated with the appropriate secondary antibody (anti-rabbit or anti-mouse IgG) for 2 h at room temperature. The immune-labeled proteins were visualized using enhanced chemiluminescence (ECL). β-actin was used as an internal control for mammalian cell proteins.

### 4.7. Subcellular Fractionation

Cytosolic and nuclear fractions of protein were fractionated using an ab109719 Cell Fraction Kit (Abcam, Cambridge, MA, USA) according to the manufacturer’s instruction. Trypsinized AGS cells and the medium were collected and centrifuged for 5 min at 300× *g*. Collected cells were washed twice with Buffer A (washing buffer). Cells were counted and resuspended in Buffer A to 6.6 × 10^6^ cells/mL. An equal volume of Buffer B (lysis buffer for cytosol extraction) was added to the cell suspensions, mixed via pipetting, and then incubated for 7 min at RT. The cell lysate was centrifuged at 5000× *g* for 1 min at 4 °C and cytosol fraction of the cell lysates in the supernatants were collected in new tubes. The pellets were resuspended in Buffer A, mixed with an equal volume of Buffer C (lysis buffer for mitochondrial extraction), mixed via pipetting, and incubated for 10 min at RT. The cell lysate was centrifuged at 5000× *g* for 1 min at 4 °C, and the supernatant was removed. The remaining pellet containing the nuclear fraction of the protein was resuspended in Buffer A.

### 4.8. Confocal Microscopy

AGS cells (2 × 10^5^ cells/well) were grown on cover slips for 24 h. The cells were then infected with 100 MOI of *H. pylori* ATCC 49503 strain and treated with 5 μM menadione for 12 h. The cells were washed twice with PBS and fixed with 1 mL of 2% paraformaldehyde for 10 min. The fixed cells were washed three times with PBS. A blocking step with 5% bovine serum albumin (BSA) for 1 h was preceded before incubation with a primary antibody. The primary antibody (anti-NF-κB antibody) was diluted in 1% BSA-PBS and treated for 3 h. The cells were washed three times with PBS and incubated with an FITC-labeled anti-mouse antibody diluted in 1% BSA for 1 h. After incubation, the cells were washed three times with PBS, treated with one drop of DAPI stain solution, and mounted on slides. The prepared slides were observed using a laser confocal scanning microscope (LSM 510, Zeiss, Heidenheim, Germany)

## 5. Conclusions

*H. pylori* colonization should be eradicated in patients with peptic ulceration to help ulcer healing and prevent ulcer recurrence. Although the success of the treatment depends on several factors such as smoking or patient compliance, antibiotic resistance is one of the main factors affecting *H. pylori* eradication. Thus development of a new therapeutic or supportive agent for eradication of *H. pylori* is necessary. We demonstrated the anti-bacterial and anti-inflammatory mechanism of menadione against *H. pylori*. To summarize, we report:

1) menadione had anti-*H. pylori* effect.

2) menadione suppressed translocation of CagA and VacA to AGS cells by decreased transcription of T4SS components involved in CagA injection and SecA involved in VacA secretion.

3) menadione inhibited NF-κB activation induced by *H. pylori* and thereby reduced the expression of pro-inflammatory cytokines including IL-1β, IL-6, IL-8, and TNF-α.

Consequently, our results suggest that menadione is a candidate of a supportive agent to treat inflammation induced by *H. pylori* as well as to eradicate *H. pylori* infection.

## Figures and Tables

**Figure 1 ijms-20-01169-f001:**
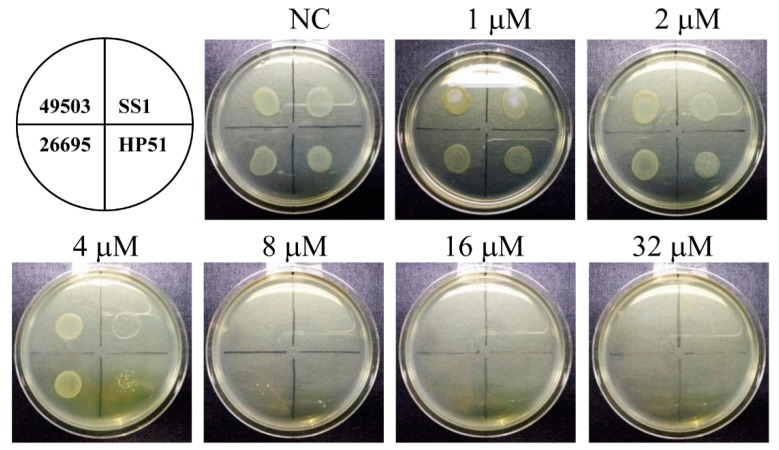
Determination of the MIC of menadione against *H. pylori* reference strains by agar dilution. Four *H. pylori* strains (ATCC 49503, SS1, ATCC 26695, and HP51) were grown on Mueller–Hinton agar containing indicated concentrations of menadione (1, 2, 4, 8, 16, and 32 μM). MIC of menadione against *H. pylori* was determined after 72 h of incubation.

**Figure 2 ijms-20-01169-f002:**
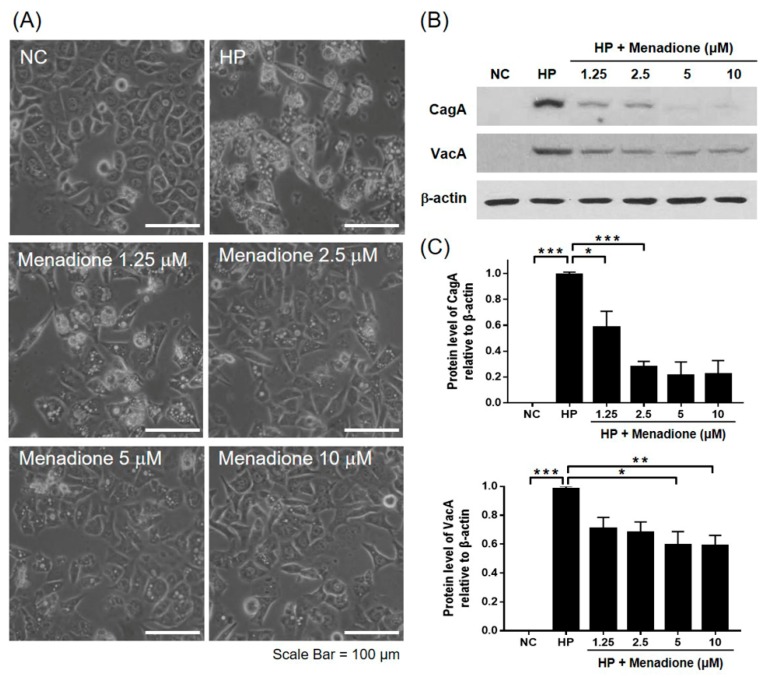
Inhibitory effects of menadione on CagA and VacA translocation to the gastric cell line and morphological changes of the gastric cell line by *H. pylori*. AGS cells were infected with *H. pylori* (200 MOI) and treated with indicated concentrations of menadione (1.25, 2.5, 5, and 10 μM) for 48 h. (**A**) After incubation, morphological changes were observed by an inverted microscope (×200) and compared to the normal control group (NC) and the *H. pylori*-infected control group (HP). (**B**) The cell lysates were subjected to Western blotting to detect CagA and VacA proteins. β-actin was used as an internal control. (**C**) Densities of the Western blotting bands were analyzed with ImageLab software. The experiments were conducted in triplicate, and the results were evaluated by a Student’s *t*-test (* *p* < 0.05, ** *p* < 0.01, and *** *p* < 0.001).

**Figure 3 ijms-20-01169-f003:**
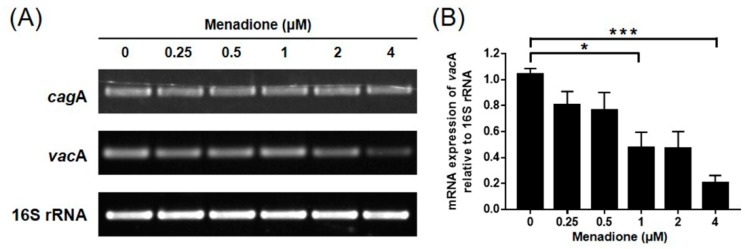
mRNA level of CagA and VacA in *H. pylori* treated with menadione. *H. pylori* specimens were treated with indicated concentrations of menadione (0.25, 0.5, 1, 2, and 4 μM) for 24 h and RNA was extracted. (**A**) Collected RNA was subjected to RT-PCR to detect mRNA expression level of *cag*A and *vac*A. Expression of 16S rRNA was used as an internal control. (**B**) Densities of the PCR bands were analyzed with ImageLab software. The experiments were conducted in triplicate, and the results were evaluated by a Student’s *t*-test (* *p*< 0.05 and *** *p* < 0.001).

**Figure 4 ijms-20-01169-f004:**
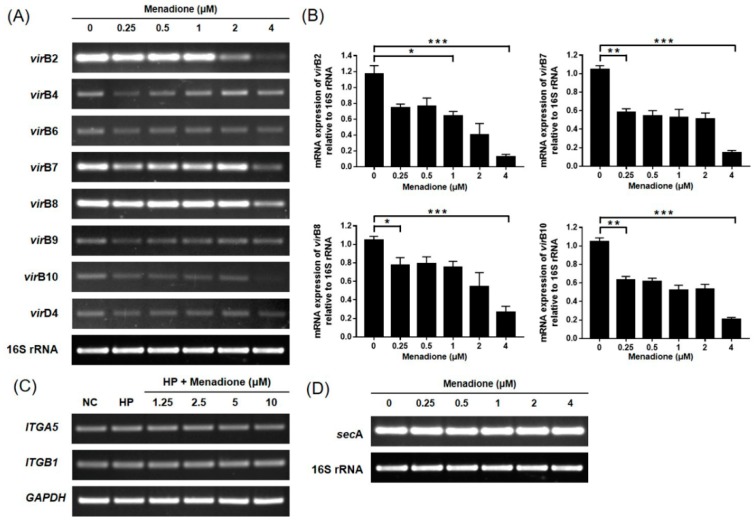
Expression of T4SS components and *sec*A in *H. pylori* treated with menadione. (**A**) *H. pylori* was treated with indicated concentrations of menadione (0.25, 0.5, 1, 2, and 4 μM) for 24 h, and RNA was extracted. Collected RNA was subjected to RT-PCR to detect the mRNA expression level of T4SS components (*vir*B2, *vir*B4, *vir*B5, *vir*B6, *vir*B7, *vir*B8, *vir*B9, *vir*B10, and *vir*D4). (**B**) Densities of the PCR bands were analyzed with ImageLab software. The experiments were conducted in triplicate, and the results were evaluated by a Student’s *t*-test (* *p* < 0.05, ** *p* < 0.01, and *** *p* < 0.001). (**C**) AGS cells were infected with *H. pylori* (200 MOI) and treated with indicated concentrations of menadione (1.25, 2.5, 5, and 10 μM) for 48 h. After incubation, RNA was extracted from the cells and subjected to RT-PCR using specific primers for integrin α_5_ and β_1_. GAPDH was used as an internal control. (**D**) *H. pylori* was treated as in (A). The mRNA expression level of *sec*A was observed by RT-PCR. Expression of 16S rRNA was used as an internal control.

**Figure 5 ijms-20-01169-f005:**
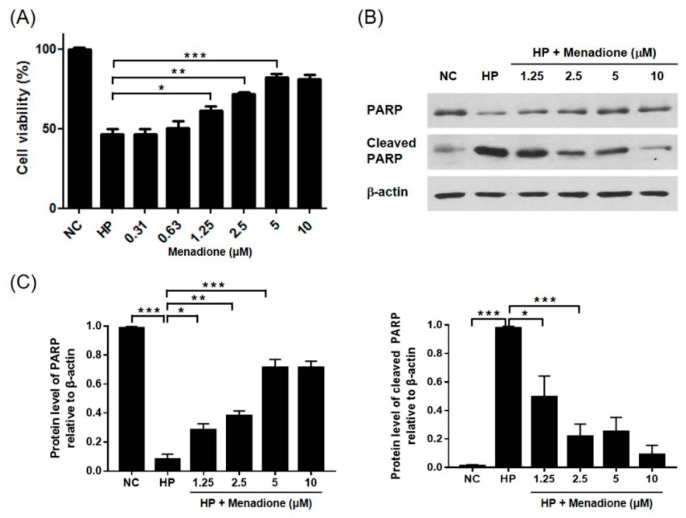
Inhibitory effect of menadione on the *H. pylori* induced cell death of gastric cells. AGS cells were infected with *H. pylori* (200 MOI) and treated with indicated concentrations of menadione for 48 h. (**A**) After incubation, cell viability was measured by the WST assay. (**B**) The cell lysates were collected to conduct Western blotting to detect full-length PARP (116 kDa) and cleaved PARP (89 kDa). β-actin was used as an internal control. (**C**) Densities of the Western blotting bands were analyzed with ImageLab software. The experiments were conducted in triplicate, and the results were evaluated by a Student’s *t*-test (* *p* < 0.05, ** *p* < 0.01, and *** *p* < 0.001).

**Figure 6 ijms-20-01169-f006:**
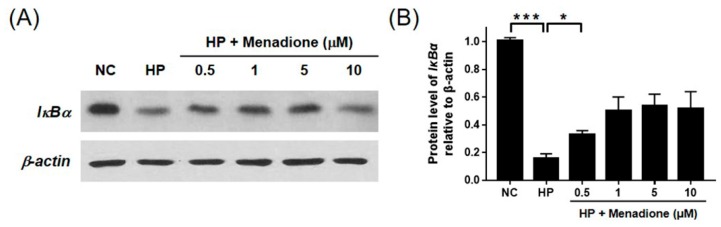
Western blotting of IκBα in AGS cells infected with *H. pylori* and treated with menadione. AGS cells were infected with *H. pylori* (200 MOI) and treated with indicated concentrations of menadione for 12 h. (**A**) After incubation, cell lysates were subjected to Western blotting to detect IκBα. β-actin was used as an internal control. (**B**) Densities of the Western blotting bands were analyzed with ImageLab software. The experiments were conducted in triplicate, and the results were evaluated by a Student’s *t*-test (* *p* < 0.05 and *** *p* < 0.001).

**Figure 7 ijms-20-01169-f007:**
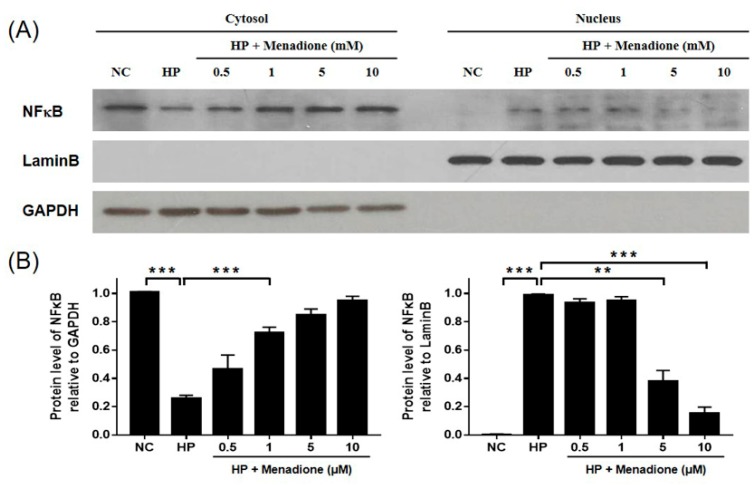
Western blotting of NF-κB in the cytosolic and nuclear fractions in AGS cells infected with *H. pylori* and treated with menadione. AGS cells were infected with *H. pylori* (200 MOI) and treated with indicated concentrations of menadione for 12 h. (**A**) After incubation, cell lysates were separated into cytosolic and nuclear fractions, then subjected to Western blotting for NF-κB. Lamin B was used as an internal control for nuclear fraction and GAPDH was used as an internal control for cytosolic fraction. (**B**) Densities of the Western blotting bands were analyzed with ImageLab software. The experiments were conducted in triplicate, and the results were evaluated by a Student’s *t*-test (** *p* < 0.01 and *** *p* < 0.001).

**Figure 8 ijms-20-01169-f008:**
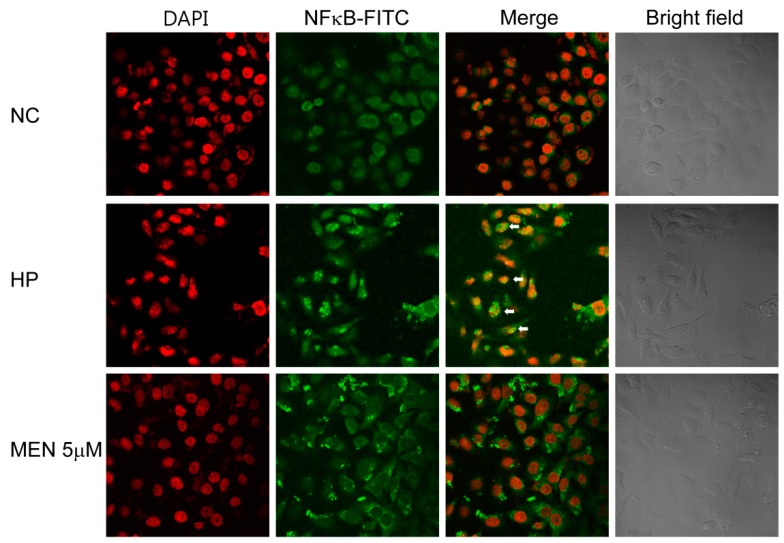
Confocal microscopy of FITC-labeled NF-κB in AGS cells infected with *H. pylori* treated with menadione. AGS cells were infected with *H. pylori* (200 MOI) and treated with 5 μM of menadione for 12 h. After incubation, NF-κB proteins in the cells were stained with mouse anti-NF-κB IgG and FITC-labeled secondary anti-mouse IgG, and the cell nucleus was selectively stained with DAPI. Images of the stained cells were then captured by confocal microscopy (×400). The white arrow indicates the nuclear localization of NF-κB.

**Figure 9 ijms-20-01169-f009:**
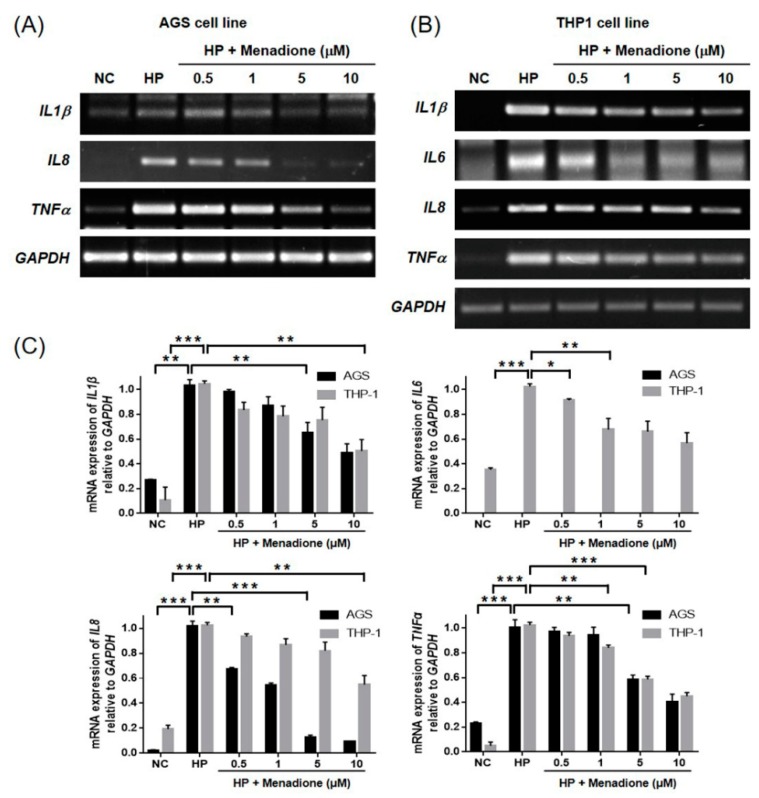
Inhibitory effects of menadione on the expression of IL-1β, IL-6, IL-8, and TNF-α in AGS cells infected with *H. pylori*. (**A**) AGS cells were infected with *H. pylori* (200 MOI) and treated with indicated concentrations of menadione for 12 h. After incubation, RNA was extracted from the cells and subjected to RT-PCR by using specific primers for IL-1β, IL-8, and TNF-α. GAPDH was used as an internal control. (**B**) THP-1 cells were activated by PMA for 48 h, and the cells were then infected with *H. pylori* (200 MOI) and treated with indicated concentrations of menadione for 12 h. After incubation, RNA was extracted from the cells and subjected to RT-PCR by using specific primers for IL-1β, IL-6, IL-8, and TNF-α. GAPDH was used as an internal control. (**C**) Densities of the PCR bands were analyzed with ImageLab software. The experiments were conducted in triplicate, and the results were evaluated by a Student’s *t*-test (* *p* < 0.05, ** *p* < 0.01 and *** *p* < 0.001).

**Table 1 ijms-20-01169-t001:** MIC of menadione on *H. pylori* clinical isolates.

Menadione Concentration (M)	Number of Strains (*n* = 38)
1	1 (2.6%)
2	4 (10.5%)
4	8 (21.1%)
8	22 (57.9%)
16	2 (5.3%)
32	1 (2.6%)

**Table 2 ijms-20-01169-t002:** List of primer sequences and PCR conditions for RT-PCR.

Primers	Sequences (5′–3′)		Product Length (bp)	Annealing Temperature (°C)	Cycles	Reference *
	Forward	Reverse				
16s rRNA	TGCAGCTAACGCATTAAGCATC	CATTCTGGCTTCAGTGTAACG	642	52	16	[63]
CagA	TGGCAGTGGGTTAGTCATA	CCTGTGAGTTGGTCTTCTTGT	278	45	35	[64]
VacA	AAACGACAAGAAAGAGATCAGT	CCAGCAAAAGGCCCATCAA	291	57	22	
VirB2	CAGTCGCCTGACCTCTTTTGA	CGGTCACCAGTCCTGCAAC	156	62	25	[65]
VirB4	GTTATAGGGGCAACCGGAAG	TTGAACGCGTCATTCAAAGC	449	62	37	
VirB5	TACAAGCGTCTGTGAAGCAG	GACCAACCAACAAGTGCTCA	436	62	29	
VirB6	CCTCAACACCGCCTTTGGTA	TAGCCGCTAGCAATCTGGTG	225	62	25	
VirB7	GATTACGCTCATAGGCGATGC	TGGCTGACTTCCTTGCAACA	202	62	25	
VirB8	GTTGATCCTTGCGATCCCTCA	CGCCGCTGTAACGAGTATTG	218	62	25	
VirB9	GCATGTCCTCTAGTCGTTCCA	TATCGTAGATGCGCCTGACC	269	62	25	
VirB10	TCCACTTCATCAGCTTGTCG	CTAACGACAGAGCGGCTATC	361	62	31	
VirD4	CCGCAAGTTTCCATAGTGTC	GCGAGTTGGGAAACTGAAGA	263	62	25	
SecA	AAAAATTTGACGCTGTGATCC	CCCCCAAGCTCCTTAATTTC	274	47	27	[66]
ITGA5	GTGACTACTTTGCCGTGAAC	AGTCGCTTACTGGGAATAGC	276	60	25	
ITGB1	GAGAATCCAGAGTGTCCCAC	ACAGTTGTTACGGCACTCTT	215	60	21	
GAPDH	CGGGAAGCTTGTCATCAATGG	GGCAGTGATGGCATGGACTG	349	55	20	[67]

* The primers without reference are designed in this study.

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
