# Peer review of "Inhibitory Effects of Menadione on Helicobacter pylori Growth and Helicobacter pylori-Induced Inflammation via NF-κB Inhibition"

_ijms, 2019, doi:10.3390/ijms20051169_

Reviewer 1 Report

Respected,

the paper is well written and intersting.

Because of the complexity of the reported data, I suggest to add a Conclusion section to give a complete and comprehensive overview of practical aspects of your study.

Kind regards

Author Response

Response to the reviewer

Point 1: Because of the complexity of the reported data, I suggest to add a Conclusion section to give a complete and comprehensive overview of practical aspects of your study.
Response 1: Thank you for your thoughtful comments to improve the manuscript. As suggested by reviewer, we have added a Conclusion section as follows:

Conclusion (Page 11, line 353) – “We confirmed the anti-bacterial activities of menadione against H. pylori at similar concentration to antibiotics for H. pylori eradication. Furthermore, we demonstrated that menadione had anti-inflammatory effects by decreasing injection of virulence factors into host cells.”

Reviewer 2 Report

In this manuscript, Lee et al. aim to investigate the effect of menadione, as a synthetic form of vitamin K, on major virulence factors of H. pylori inflammation mediated by this microorganism. Using various methodological approaches, they have revealed that menadione inhibits H. pylori growth. Also, it has down-regulated expression of vacA in H. pylori associated with decrease in translocation of VacA into a gastric adenocarcinoma cell line. They have further demonstrated a pro-apoptotic effect of menadione on these cells. Interestingly, menadione has suppressed inhibited NF-kB activation leading to inhibition of inflammatory response. The authors believe that their in vitro data shows that menadione plays both anti-bacterial and anti-inflammatory roles in H. pylori infection.

I have the following questions and comments to be addressed prior to acceptance for publication:

1)    Lack of positive control throughout the experiments. It is necessary to compare the effect of menadione with antibiotics (at least one of them) clinically administered to treat peptic ulcer. The potential synergistic effect of menadione and these antibiotics could be examined.

2)    Quality of imaging data. There is a strong background (green) staining in Figure 2 and 5. The background must be dark enough to convince the audience. A bigger magnification of these images is highly recommended. In addition, scale bars must be added to imaging data.

3)      What is the effect of menadione alone (without H. pylori) on the expression of VacA, CagA, PARP, NF-kB, IkBa as well as pro-inflammatory cytokines and virulence factors?

4)      Why concentrations of menadione vary from experiment to experiment?

5)      Densitometric analysis. All RT-PCR and WB data must be analyzed by a densitometry software and statistically compared between the treated versus untreated groups (at least 3 replicates).

Author Response

Response to Reviewer

Point 1: Lack of positive control throughout the experiments. It is necessary to compare the effect of menadione with antibiotics (at least one of them) clinically administered to treat pectic ulcer. The potential synergistic effect of menadione and these antibiotics could be examined.
Response 1: In our previous report, we evaluated MIC of the antibiotics on the clinical H. pylori strains. As suggested, therefore, we have added sentences to the discussion section to compare the effect of menadione with antibiotics as follows:

Discussion (Page 9, line 283) – “The MIC of menadione was 8 uM (1.38 ug/mL) which was similar to the MIC of the antibiotics clinically administered to treat H. pylori infection. In our previous report, we isolated 165 H. pylori clinical strains and evaluated MIC of five antibiotics commonly used for the eradication of H. pylori. The MICs of the clarithromycin, amoxicillin, tetracycline, metronidazole and levofloxacin against the susceptible H. pylori strains were 0.008~0.125 ug/mL, 0.008~0.5 ug/mL, 0.031~2 ug/mL, 0.063~4 ug/mL, and 0.008~1 ug/mL.”

Point 2: Quality of imaging data. There is a strong background (green) staining in Figure 2 and 5. The background must be dark enough to convince the audience. A bigger magnification of these images is highly recommended. In addition, scale bars must be added to imaging data.
Response 2: As suggested, we have added scale bars and removed green staining on the background and improved the resolution of Figure 2.

Point 3: What is the effect of menadione alone (without H. pylori) on the expression of VacA, CagA, PARP, NF-kB, IkBa as well as pro-inflammatory cytokines and virulence factors?

Response 3: As your suggestion, we have evaluated the effect of menadione alone on the cells without H. pylori infection, and found that menadione alone has no effect on the level of the molecules in the range of concentration used in our study. We have shown the results in the Supplementary Figure 1 and added a Result section as follows:

Page 6, line 197 – “Menadione alone had no effect on the level of PARP in the range of concentration (Supplementary Figure 1).”

Page 7, line 233 – “AGS cells without H. pylori infections were treated with menadione, there was no variation of IkBa and NF-κB (Supplementary Figure 1).”

Point 4: Why concentrations of menadione vary from experiment to experiment?

Response 4: In our study we have shown that H. pylori growth was inhibited by 8 uM of menadione. Thus it was difficult to harvest mRNA and protein from H. pylori treated with 8 uM or higher concentration of menadione. This is why we treated menadione upto 4 uM when we harvest mRNA or protein from H. pylori. However, in the experiments with the mammalian cells infected with H. pylori, we wanted to observe the effect of menadione at the range of concentration including inhibitory concentration against the bacteria. Thus we adjusted the menadione concentration upto 10 uM when we harvest mRNA or protein from mammalian cells.

Point 5: Densitometric analysis. All RT-PCR and WB data must be analyzed by a densitometry software and statistically compared between the treated versus untreated groups (at least 3 replicates)

Response 5: As your suggestion, we have analyzed RT-PCR and WB data with densitometry software. The graphs with statistical analysis were added to each figure. Description of the graphs has been added to the figure legends as follows:

Figure 2 (Page 4, line 148) – “(C) Densities of the Western blotting bands were analyzed by ImageLab software. The experiments were conducted in triplicate and the results were evaluated by Student’s t-test (*P<0.05, **P < 0.01 and ***P<0.001).”< span="">

Figure 3 (Page 5, line 155) – “(B) Densities of the PCR bands were analyzed by ImageLab software. The experiments were conducted in triplicate and the results were evaluated by Student’s t-test (*P<0.05 and ***P<0.001).”< span="">

Figure 4 (Page 6, line 175) - “(B) Densities of the PCR bands were analyzed by ImageLab software. The experiments were conducted in triplicate and the results were evaluated by Student’s t-test (*P<0.05, **P < 0.01 and ***P<0.001).”< span="">

Figure 5 (Page 6, line 206) – “(C) Densities of the Western blotting bands were analyzed by ImageLab software. The experiments were conducted in triplicate and the results were evaluated by Student’s t-test (*P<0.05, **P < 0.01 and ***P<0.001).”< span="">

Figure 6 (Page 7, line 232) – “(B) Densities of the Western blotting bands were analyzed by ImageLab software. The experiments were conducted in triplicate and the results were evaluated by Student’s t-test (*P<0.05 and ***P<0.001).”< span="">

Figure 7 (Page 8, line 241) – “(B) Densities of the Western blotting bands were analyzed by ImageLab software. The experiments were conducted in triplicate and the results were evaluated by Student’s t-test (**P < 0.01 and ***P<0.001).”< span="">

Figure 9 (Page 9, line 272) – “(C) Densities of the PCR bands were analyzed by ImageLab software. The experiments were conducted in triplicate and the results were evaluated by Student’s t-test (*P<0.05, **P < 0.01 and ***P<0.001).”< span="">

Round  2

Reviewer 2 Report

They have addressed the questions I was asking. The manuscript could be accepted as the current revision.